# Hypoalbuminemia and Pharmacokinetics: When the Misunderstanding of a Fundamental Concept Leads to Repeated Errors over Decades

**DOI:** 10.3390/antibiotics12030515

**Published:** 2023-03-04

**Authors:** Peggy Gandia, Sarah Decheiver, Manon Picard, Romain Guilhaumou, Sarah Baklouti, Didier Concordet

**Affiliations:** 1Laboratoire de Pharmacocinétique et Toxicologie Clinique, Institut Fédératif de Biologie, CHU de Toulouse, 330 avenue de Grande-Bretagne, CEDEX 9, 31059 Toulouse, France; 2INTHERES, Université de Toulouse, UMR 1436, INRAE, ENVT, 23 Chemin des Capelles, CEDEX 3, 31076 Toulouse, France; 3AP-HM Hôpital de la Timone, Service de Pharmacologie Clinique et Pharmacovigilance, 264 rue Saint Pierre, 13005 Marseille, France

**Keywords:** hypoalbuminemia, unbound/free concentration, misinterpretation, therapeutic drug monitoring

## Abstract

Surprisingly, misinterpretation of the influence of hypoalbuminemia on pharmacokinetics and the clinical effects of drugs seems to be a current problem, even though hypoalbuminemia has no impact on the pharmacologically active exposure. Exceptions to this fact are highly protein-bound anaesthetics with high elimination capacity (i.e., <5 drugs on the market). To assess the frequency of misinterpretation of the influence of hypoalbuminemia on pharmacokinetics and the clinical effects of drugs between 1975 and 2021, a PubMed literature review was conducted. Each paragraph on albumin binding was classified as correct, ambiguous or incorrect, creating two acceptable categories: (1) content without any errors, and (2) content containing some incorrect and/or ambiguous statements. The analyses of these articles showed that fewer than 11% of articles contained no interpretation errors. In order to contain this misinterpretation, several measures are proposed: (1) Make the message accessible to a wide audience by offering a simplified and didactic video representation of the lack of impact of albumin binding to drugs. (2) Precise terminology (unbound/free form/concentration) should be used for highly bound drugs. (3) Unbound/free forms should be systematically quantified for highly plasma protein bound drugs for clinical trials as well as for therapeutic drug monitoring.

## Take-home message:

The misinterpretation of the influence of hypoalbuminemia on drug pharmacokinetics and clinical effects is still a problem in 2022.Two major levers can be used to reduce this endemic error: (i) improve the way in which the protein-binding concept is taught to medical/pharmacy students; and (ii) systematically quantify the unbound/free form for drugs highly bound to plasma proteins for clinical trials as well as for therapeutic drug monitoring.

## 1. Introduction

Moderate and severe hypoalbuminemia is frequently reported in malnourished patients and in individuals presenting systemic inflammatory syndrome, chronic gut disease or nephrotic syndrome [1,2,3]. In these cases, and given that albumin is a key drug binding protein, some pharmacists and clinicians have questioned the impact of variations in albuminemia on circulating drug concentrations and thus on drug efficacy, with a particular focus on highly bound drugs (i.e., >90%).

Indeed, in their minds and rather confusedly, hypoalbuminemia tends to vary the “effective” concentration requiring dose adjustments in order to maintain efficacy and avoid toxicity. For many healthcare professionals, the underlying mechanism is the detachment of the drug from its albumin-binding sites, thus leading to an increase in the unbound/free form. This conjures up the mental image of the body operating as a tank filled with stagnant water. The detached drug remains in the tank, thus leading to increased unbound/free concentration without any variation in the total concentration, similar to the in vitro investigation of hypoalbuminemia. However, this mental image is incorrect and, consequently, leads to misinterpretation of the impact of hypoalbuminemia on pharmacokinetics. Indeed, when a patient presents hypoalbuminemia, the bound concentration decreases while the unbound/free concentration remains unchanged [4,5,6,7,8,9,10,11]. This reduces the measured concentration (i.e., sum of unbound/free and bound forms). As a consequence, the unbound/free fraction (i.e., ratio of the unbound/free concentration on the total concentration) increases due to a decreased total concentration, while the unbound/free concentration remains unchanged.

From a mathematical point of view, this is described explicitly by the following equations that are also reported in Toutain et al. [4]. To facilitate comprehension, we consider all concentrations at the steady state:(1)$$C_{tot}=C_{free}+C_{bound}$$
where $C_{free}$ and $C_{bound}$ are the free/unbound and bound concentrations, respectively. For a drug with a single family of binding sites (binding sites characterized by the same affinity constant $k_{a}$), $C_{bound}$ is given by the general equation:(2)$$C_{bound}=\frac{B_{max}\times C_{free}}{K_{D}+C_{free}}$$
where $B_{max}$ is the maximal binding capacity (related to the molar concentration of the binding protein), and $K_{D}$ is the equilibrium dissociation constant. Incorporating Equation (2) into Equation (1) gives:(3)$$C_{tot}=C_{free}+\frac{B_{max}\times C_{free}}{K_{D}+C_{free}}$$

Thus, when hypoalbuminemia occurs, one can see that: $C_{bound}=\frac{B_{max}\times C_{free}}{K_{D}+C_{free}}$ decreases because $B_{max}$ decreases without affecting $C_{free}$.

For the same reason, hypoalbuminemia causes $C_{tot}=C_{free}+\frac{B_{max}\times C_{free}}{K_{D}+C_{free}}$ to decrease because $B_{max}$ decreases without affecting $C_{free}$.

In the same way, the impact of hypoalbuminemia on $f_{u}=\frac{C_{free}}{C_{tot}}=\frac{C_{free}}{C_{bound}+C_{free}}$ can be deduced from the previous remarks: in the case of hypoalbuminemia, $C_{bound}$ (and consequently $C_{tot})$ decreases, which increases $f_{u}$, while the unbound/free concentration remains unchanged.

In summary, because the free fraction ($f_{u}$) changes with $B_{max}$, its interpretation is misleading.

More recently, T’Jollyn et al. [12] published the kinetic profile of unbound/free and total concentrations when hypoalbuminemia occurs. Their simulations once again confirm the explanations presented above.

Even if there are few (i.e., less than 5 marketed drugs) exceptions to the rule (i.e., highly bound drugs with high elimination capacities) [13,14], there is no room to generalize these exceptions. Thus, the selection criterion applied in our study excluded drugs with a high elimination capacity. In this way, all the included drugs had only a low elimination capacity as there were no anti-infectious drugs with medium elimination capacity. Elimination capacity is characterized by the extraction ratio (ER) determined by the ratio of hepatic/renal clearance to hepatic/renal blood flow. For a low elimination capacity, the ER is considered to be <0.3, whereas for high elimination capacity, ER > 0.7. 

A common source of confusion, which will not be discussed in this article, is the change in glomerular filtration rate (GFR), which controls the unbound/free concentration of renally eliminated drugs. When the GFR decreases (e.g., in renal failure) or conversely increases (e.g., in a hyperfiltrating patient), the concentration increases or decreases, respectively. Thus, a change in GFR concomitant with hypoalbuminemia leads to the mistaken belief that hypoalbuminemia is the cause of the change in unbound/free drug concentration.

Misunderstanding the protein-binding concept provides one explanation for incorrect and sometimes dangerous drug dosage adjustments.

In the 2000s, our colleagues [4,7,9], some of whom are the founders of modern pharmacology, wrote didactic research papers to explain protein binding and dispel misconceptions. The understanding of the impact of plasma protein binding in the 2020s seems still very unclear. The purpose of this paper is to assess the prevalence of this error in medical literature from the 1970s to the present day, focusing on dosing adjustments for patients presenting hypoalbuminemia. This work was focused on anti-infective drugs. Indeed, anti-infective drugs are widely prescribed and a drug dosage adjustment based on a therapeutic drug monitoring (TDM) is recommended for patients most at risk of therapeutic failure or toxicity.

## 2. Results

Flowchart in Figure 1 shows the methodology used to select PubMed articles. This methodology follows the recommendations given in [15].

Three hundred and three articles were identified from 1975 to 2021 (Figure 2) and screened.

One hundred and five articles were first selected for our global analysis. Then, only articles referring to anti-infective drugs were included (Table 1).

The list of anti-infective drug related articles and an analysis of their contents (regarding the interpretation of hypoalbuminemia on unbound/free exposure) is provided in Appendix A [16,17,18,19,20,21,22,23,24,25,26,27,28,29,30,31,32,33,34,35,36,37,38,39,40,41,42,43,44,45,46,47,48,49,50,51,52,53,54,55,56,57,58,59,60,61,62,63,64,65,66,67,68,69,70,71,72,73,74,75,76]. Thorough reading of its contents is beneficial.

An increasing number of manuscripts have been published on this topic over the last fifty years. Regardless of the period in question, fewer than 11% articles did not contain any errors in interpreting the impact of hypoalbuminemia on unbound/free drug exposure (category 1). Similarly, fewer than 43% articles did not contain entirely false statements on the impact of hypoalbuminemia on unbound/free drug exposure despite containing ambiguous statements (category 2).

All the results are contained in the Appendix A, which is over 30 pages long. Due to the length of the Appendix A, a selection of the most representative examples is presented below (Table 2 and Table 3):-***The correct explanations:***

“**A previous study indicated that although clearance of total daptomycin (CL) was affected by alterations in fu, CLu did not get affected**. In our study, fu ranged from 0.05 to 0.14 depending on the influence of serum albumin, BUN, and FBG. Regarding factors affecting fu and CL, reports on teicoplanin, which is an antimicrobial agent with a high protein binding rate, identified serum albumin and FBG. In our study, the results were similar to those of teicoplanin, and, to the best of our knowledge, this is the first report on factors that affected fu in daptomycin. **Moreover, our results suggested that because CL varies with the influence of fu, establishing the dose using total concentrations may result in an under- or overestimation**.” [17]

-
**
*The ambiguous explanations reported in five different cases:*
**


**Table 2 antibiotics-12-00515-t002:** Examples of ambiguous explanations found in the literature.

Cases	Ambiguous Sentences	Explanations
**A**	“**Calculation of the unbound concentrations, assuming 95% protein binding, may therefore result in considerable overdosing, in particular in critically ill patients with hypoalbuminaemia and renal impairment. **In the present study, the inter-individual unbound plasma fraction of flucloxacillin varied widely from 1.1% to 64.7%, showing a substantially higher median value (11%) than reported for healthy individuals (5%).” [32]	Combining the impact of hypoalbuminemia and a change in kidney/liver and drug-drug interactions in the same sentence makes it difficult to identify clinical repercussions. Indeed, hypoalbuminemia does not alter unbound/free concentrations and cannot have any clinical impact. On the contrary, a defective excretory system reduces unbound/free/intrinsic clearance, leading to greater unbound/free exposure and, therefore, impacts the clinical effect of the drug. Moreover, when two drugs (A and B) are administered concomitantly, drug A can modify the protein-binding of drug B without altering the latter’s unbound/free concentration. However, it cannot be ruled out that drug A will also decrease the clearance of drug B, thereby potentially increasing the unbound/free exposure (i.e., concentrations) of drug B. Two mechanisms are, therefore, involved in the drug A–drug B interaction.
**B**	“The **binding of beta-lactams to albumin and plasma proteins determines the free fraction, which is the biologically active fraction that diffuses across biological membranes to tissues. The free fraction is also the fraction that is eliminated by renal and liver clearance.** When **plasma protein amount decreases, the capacity of beta-lactams to bind to protein decreases and beta-lactam-free fraction increases**. Previous studies have shown that the binding of beta-lactams to plasma proteins in ICU patients is highly variable and is more altered for antibiotics highly bound to plasma proteins in conditions of homeostasis (e.g., ceftriaxone, cefazolin, or ertapenem). As a result, **plasma concentration of beta-lactam antibiotics may be lowered and more unpredictable in patients with severe hypoalbuminemia.**” [38]	There is confusion of the terms, “unbound/free form” and “unbound/free fraction” as in the sentence “the unbound/free fraction is the pharmacologically active fraction”. A number (i.e., unbound/free fraction) cannot be pharmacologically active, unlike the drug form. Indeed, the unbound/free form of the drug is the pharmacologically active form regardless of the albuminemia and unbound/free fraction. Moreover, the value of the unbound/free fraction is misleading because it increases with the hypoalbuminemia and thus incorrectly suggests that the unbound/free concentration also rises, thereby suggesting an enhanced clinical effect with the drug in question.
**C**	“SAFE Study Investigator reported that approximately 40% of critically ill patients presented with hypoalbuminemia, because cefoperazone, meropenem, and imipenem are highly bound to albumin, which could **increase the unbound fraction significantly**. **Therefore, various pathological characteristics in critically ill patients may induce a wide discrepancy in the unbound fraction concentrations**.” [34]	Use of the term, “unbound/free fraction” instead of “unbound/free concentration” is confusing because the unbound/free fraction changes with hypoalbuminemia, while the unbound/free concentration is unchanged. Consequently, the variation in terms of the unbound/free fraction sheds no light on the clinical effect of the drug.
**D**	“Probably due to this dramatic increase in the ƒu, our patients exhibit a higher ceftriaxone CL than healthy volunteers or critically ill patients with sepsis, septic shock, and different degrees of renal function, a CL that is dependent on albumin concentration and weight based on the results of the population PK model. **Surprisingly, in spite of the augmented CL, dosing simulations show that, for an MIC ≤ 2 mg/L** (the clinical breakpoint for susceptibility to ceftriaxone), **a dose of 1000 mg q24 h maintains unbound ceftriaxone concentrations for a 100% of the dosing interval above the MIC regardless of albumin concentration and body weight**.” [31]	Using expressions such as “mostly, usually, often, almost, potentially, may, in several cases, surprisingly…” suggests that the fact is only observed in specific cases, as indicated in the sentence, “Serum albumin concentrations may not affect unbound/free concentrations”, whereas this is always the case.
**E**	“In addition, **low serum albumin concentration is frequently observed in ICU patients, leading to an increase in the free fraction of the beta-lactams highly bound to plasma proteins, such as cefazoline, ceftriaxone, or ertapenem. Thus, ****hypoalbuminemia may lead to increased Vd and tissue penetration, and also increased elimination, of beta-lactam antibiotics by glomerular filtration and/or metabolic clearance**. This has been particularly observed for ceftriaxone or ertapenem.” [38]	The paper gives correct conclusions on total pharmacokinetic parameters of no clinical consequence. For example, hypoalbuminemia implies an increase in total clearance without affecting the unbound/free/intrinsic clearance that controls unbound/free exposure. This information misleadingly suggests to the reader that the unbound/free concentration is lower because of increased total clearance. In another example, hypoalbuminemia implies an increase in the total volume of distribution without affecting the unbound/free volume of distribution. This information incorrectly suggests to the reader that the kinetic profile of the unbound/free concentration is modified because of the increase in the total volume of distribution.

TZP: piperacillin/tazobactam; MER: meropenem; PK: pharmacokinetic; fu: unbound/free fraction; CL: clearance; MIC: minimum inhibitory concentration; ICU: intensive care unit; Vd: distribution volume.

-
**
*The incorrect explanations:*
**


**Table 3 antibiotics-12-00515-t003:** Examples of incorrect explanations found in the literature.

Incorrect Sentences	Explanations
“We recognise that our case series is limited and that the study design was retrospective and monocentric. Additionally, **only total cefiderocol concentrations were measured, thus potential variability in protein binding commonly encountered in critically ill patients could impact on cefiderocol free levels**.” [58]	Variation in unbound concentration occurs over a very short duration (probablyseconds or minutes). However, unbound/free concentration returns to the baseline level, while unbound/free fraction (fu) increases. This increase in fu leads to an increase in the clearance and the volume of distribution for total concentration but not for unbound/free concentration.

PK: pharmacokinetic.

Misunderstanding the protein-binding concept is also palpable when one examines the way in which this concept is discussed by the same research team through its various publications:-None of the selected articles contained exclusively true assertions for a specific author;-Some authors wrote both correct and ambiguous sentences in different articles ([20]/[27]; [22]/[47]; [23]/[41]);-Some authors have remained consistent by repeating the same incorrect message over time, suggesting that this concept was not taught to them properly at medical school (references of two articles with wrong assertions: [64]/[74]; [66]/[67]; [72]/[73]).-Some authors state in articles published a few months apart (or even in the same article), facts that contradict each other (references of two articles with correct and incorrect assertions, respectively: [21]/[59]; [20]/[59]; [19]/[41]; [19]/[39]; [22]/[25]; [23]/[66]; [21]/[66]; [21]/[21]; [25]/[25]).

## 3. Discussion and Conclusions

The impact of hypoalbuminemia on pharmacokinetics is largely misunderstood by healthcare professionals. This is an observation reflected in the increasing number of manuscripts published on this topic over the last fifty years that contain interpretation errors. Regardless of the period in question, around 89% of articles contain ambiguous and/or false ([19 + 35/61 articles] × 100) statements on the impact of hypoalbuminemia on unbound/free drug exposure. Surprisingly, and despite the fact that Benet’s, Rolan’s and Toutain’s articles [4,7,9] have been cited on numerous occasions (517, 310 and 110 times, respectively), the percentage of articles containing misleading or confusing information did not decrease following their publication in the early 2000’s. The reasons are not totally clear but some of them seem obvious: (i) some research units/laboratories do not technically have the possibility of determining the unbound/free form, whereas total concentration is easily accessible with current analytical tools; (ii) the level of pharmacokinetic knowledge is not harmonised between the different teams publishing on the impact of hypoalbuminemia on drug protein binding, leading to confusion between “unbound/free concentration” and “unbound/free fraction”; and (iii) the third argument is certainly linked to a lack of harmonisation of teaching on the influence of hypoalbuminemia on drug protein binding between different universities in the same country, or even between different specialisations (pharmacy, medicine, etc.). These accumulated reasons, combined with the need for research units to publish, lead to the publication of results that continue to perpetuate errors in the pharmacokinetic interpretation of hypoalbuminemia.

The main consequence of a poor understanding of drug binding to plasma proteins is the overexposure of patients with hypoalbuminemia. Indeed, in case of hypoalbuminemia, total concentration (i.e., concentration measured by the pharmacology-toxicology laboratories in hospital practices) decreases, while free/unbound concentration remains unchanged, leading clinicians to increase the patient dose in order to reach a target based on total concentration. This is typically the error encountered today with dalbavancin, ceftriaxone, cefazolin, cloxacillin and ertapenem TDM, especially because a patient’s hypoalbuminemia is not systematically examined when interpreting total concentration.

The concept of protein binding applies to all plasma proteins. Thus, a misunderstanding can lead to underexposure. For example, this kind of underexposure was observed with lopinavir used in the early weeks of the COVID-19 pandemic. As lopinavir is highly bound to alpha-1-acid glycoprotein (AAG), an increase in AAG induced by a cytokine storm resulted in an increase in bound and thus total concentrations without a change in free/unbound concentration [77]. This led medical teams to apply lower doses due to total concentrations four to five times those reported in HIV+ patients treated with lopinavir.

Overexposures are of no consequence for drugs with a wide therapeutic range (e.g., isavuconazole, posaconazole). On the other hand, they can induce toxicity for drugs with a narrow therapeutic range (e.g., valproic acid).

Conversely, the consequence of under-exposure is the lack of efficacy. However, this situation is not expected in cases of hypoalbuminemia.

A visual dynamic representation of the influence of hypoalbuminemia on bound and total concentrations is available in our sound video (https://www.youtube.com/watch?v=4SXY8YyRbQo).

As mentioned previously, there are few exceptions to the rule (i.e., bupivacaine, propofol, sufentanil). These drugs are highly bound drugs with high elimination capacities [13,14]. For these drugs, which are always administered intravenously, hypoalbuminemia leads to an increase in unbound/free concentrations, whereas the total concentrations remain unchanged. As there is no TDM for this kind of drugs (as it is not suitable), it is essential (i) not to generalize these exceptions and (ii) to keep in mind the correct interpretation (i.e., there is no impact of hypoalbuminemia on unbound concentrations).

In summary, the most important consequences of misinterpreting hypoalbuminemia are for drugs that are highly related and have a narrow therapeutic range. These drugs are mainly used in hospitals and, therefore, concern a limited number of patients; the consequences of an iatrogenic overexposure in general practice would probably be much more serious.

To conclude, it is possible that in the future, healthcare professionals may no longer confuse unbound/free drug fraction and unbound/free drug concentration to adapt the dosage of antibiotics in patients with hypoalbuminemia. With this aim, the solution is obvious and warrants a two-fold approach: (i) to improve the way in which the protein-binding concept is taught at medical school, in particular by prohibiting the use of the “unbound/free fraction” terminology for highly bound drugs and by using the “unbound/free concentration instead” terminology; and (ii) to systematically quantify the unbound/free form for drugs highly bound to plasma proteins in both clinical trials and TDM.

Advancements in technology, both in the analytical and pedagogical fields, (e.g., ChatGPT) should improve the situation.

## 4. Methods

We conducted a PubMed literature review from January 1st, 1975 to December 31th, 2021. We looked for all articles including an interpretation of free/unbound fraction by using the following key terms in our search: (patients) AND (hypoalbuminemia) AND (“protein binding”) OR (“critically ill”) AND (“free fraction”) OR (“unbound fraction”) OR (“free concentration”) OR (“unbound concentration”) OR (“free drug concentration”) OR (“unbound drug concentration”) OR (“free concentrations”) OR (“unbound concentrations”) OR (“free drug concentrations”) OR (“unbound drug concentrations”) OR (“free plasma”) OR (“unbound plasma”) OR (“free serum”) OR (“unbound serum”) OR (“therapeutic drug monitoring” AND “albuminemia”).

Some articles did not fall into the scope of this study based on this initial sequence. We excluded (a) in vitro studies, (b) in vivo studies performed in animals, (c) articles on highly bound drugs with high elimination capacities, (d) articles on endogenous drugs and articles on endogenous substances with retrocontrols (e.g., hormones), (e) articles written in a language other than English, (f) articles with no full text available, (g) duplicates/triplicates (i.e., articles with the same title by the same team published in different journals), and (h) articles not covering protein binding, even if the keywords highlighted them in PubMed. The remaining articles were those described as “global articles” in this review. Then, a last selection was made based on the mention of anti-infective drugs among the global articles. Indeed, anti-infective drugs are widely prescribed. In hospitals, for most of them, drug dosage adjustment is based on the therapeutic drug monitoring that implies an assay of total concentration.

For the evaluation process, we classified the contents of an article into three categories:-*False*: The authors state that hypoalbuminemia leads to higher unbound/free concentrations and/or higher/lower unbound/free/intrinsic clearance and/or higher/lower unbound/free volume of distribution.-*Ambiguous*: The paper suggests, without stating it explicitly, that hypoalbuminemia can impact unbound/free concentration. Various examples of ambiguous sentences are provided below:**Case A:** Combining the impact of hypoalbuminemia and a change in kidney/liver and drug–drug interactions in the same sentence/section makes it difficult to identify clinical repercussions. Indeed, hypoalbuminemia does not alter unbound/free concentrations and cannot have any clinical impact. On the contrary, a defective excretory system reduces unbound/free/intrinsic clearance leading to greater unbound/free exposure and, therefore, impacts the clinical effect of the drug. Moreover, when two drugs (A and B) are administered concomitantly, drug A can modify the protein-binding of drug B without altering the latter’s unbound/free concentration. However, it cannot be ruled out that drug A will also decrease the clearance of drug B, thereby potentially increasing the unbound/free exposure (i.e., concentrations) of drug B. Two mechanisms are, therefore, involved in the drug A–drug B interaction.**Case B:** Confusion of the terms, “unbound/free form” and “unbound/free fraction” as in the sentence “the unbound/free fraction is the pharmacologically active fraction”. A number (i.e., unbound/free fraction) cannot be pharmacologically active, unlike the drug form. Indeed, the unbound/free form of the drug is the pharmacologically active form regardless of the albuminemia and unbound/free fraction. Moreover, the value of the unbound/free fraction is misleading because it increases with the hypoalbuminemia and thus incorrectly suggests that the unbound/free concentration also rises, thereby suggesting an enhanced clinical effect with the drug in question.**Case C:** Use of the term, “unbound/free fraction” instead of “unbound/free concentration” is confusing because the unbound/free fraction changes with hypoalbuminemia, whereas the unbound/free concentration is unchanged. Consequently, the variation in terms of the unbound/free fraction sheds no light on the clinical effect of the drug.**Case D:** Using expressions such as “mostly, usually, often, almost, potentially, may, in several cases, surprisingly…” suggests that the fact is only observed in specific cases as indicated in the sentence, “Serum albumin concentrations may not affect unbound/free concentrations”, whereas this is always the case.**Case E:** Giving correct conclusions on total pharmacokinetic parameters is of no clinical consequence. For example, hypoalbuminemia implies an increase in total clearance without affecting the unbound/free/intrinsic clearance that controls unbound/free exposure. This information misleadingly suggests to the reader that the unbound/free concentration is lower because of increased total clearance. In another example, hypoalbuminemia implies an increase in the total volume of distribution without affecting the unbound/free volume of distribution. This information incorrectly suggests to the reader that the kinetic profile of the unbound/free concentration is modified because of the increase in the total volume of distribution.-*True*: The sentence states explicitly that hypoalbuminemia has no impact on the unbound/free concentration.

Some articles make false and/or ambiguous and/or correct assertions relating to the impact of hypoalbuminemia on unbound/free drug concentrations. We consequently propose two classifications:-**Category 1:** an article is retained when it contains only correct assertions; an article is deemed unacceptable when it contains ambiguous and/or false assertions.-**Category 2:** an article is retained when it contains correct or ambiguous assertions; an article is deemed unacceptable when it contains only incorrect assertions.

To give a visual dynamic representation of the influence of hypoalbuminemia on bound and total concentrations, we created a sound video (https://www.youtube.com/watch?v=4SXY8YyRbQo). We used a single-compartment model and continuous infusion to simplify the image. However, the same mechanism applies for more complex models (n-compartments with n > 1) and intravenous/oral discontinuous drug administration. A description of the different periods of the movie is detailed below:**Period A:** From drug administration to steady state
-The black squares represent albumin molecules (1 black square = 1 albumin molecule). These squares remained fixed to make this video watchable. The red dots appearing at the top of the figure represent the drug molecules (i.e., 1 red dot = 1 drug molecule) injected into the blood (central compartment) at a constant perfusion rate (i.e., X molecules per hour);-When the drug (red dots) binds to albumin (black squares), the black squares turn green to illustrate the albumin-binding of the drug. In an attempt to simplify the presentation, we illustrated the drug bound to albumin as a static relationship throughout all of the simulation periods. There is actually a continuous “bond and detachment” process that would have been difficult to represent;-All red dots travel from the top of the figure to the bottom at the same pace. This means that the drug molecules pass through the excretory organ at the same speed, thereby mimicking intrinsic clearance. This speed does not change over time (i.e., intrinsic clearance is constant and independent of time and dose/concentration);-At the beginning of the drug perfusion stage, a few red dots reach the bottom of the figure. This shows that a few drug molecules are eliminated at the start of drug perfusion. Indeed, when the first drug molecules reach the blood, they immediately bind to albumin as the drug has a high affinity for albumin. More and more black squares gradually turn green as increasing numbers of drug molecules bind to albumin;-Due to continuous perfusion, drug accumulation in the body simultaneously leads to an increase in the number of red dots in the figure (i.e., an increase in drug concentration) until a perfusion and a steady elimination rate is reached;-Once the steady state has been reached, (i) the ratio of red dots (i.e., unbound drug molecules) to green squares (i.e., bound drug molecules) is constant, (ii) the number of red dots expelled at the bottom of the figure per unit of time is constant, and (iii) the number of red dots injected into the central compartment per unit of time is equal to the number of red dots expelled at the bottom of the figure per unit of time.
**Period B:** Severe hypoalbuminemia
-When severe hypoalbuminemia occurs (i.e., a marked decrease in albumin molecules, despite the fact that severe hypoalbuminemia takes many days to appear), a large number of green squares disappear and an equal number of red dots appear;-This increase in red dots indicates an increase in unbound/free drug molecules as well as an increase in unbound/free drug concentrations;-The excess red dots are channeled downwards at the same speed for all the dots. This means that the surplus unbound/free drug molecules are gradually eliminated by the body, with unbound/free drug concentrations being restored to the original steady state driven by both the perfusion rate and intrinsic clearance;-Peak concentrations are transient and of short duration, as highlighted by the concentration-versus-time curve of the unbound/free drug.
**Period C:** New steady state with hypoalbuminemic status
-When the peak has been reached and the number of albumin molecules has decreased, the number of bound drug molecules falls along with the overall number (bound + unbound) of drug molecules. Consequently, bound and total concentrations are lower when comparing hypoalbuminemia to the values reported at steady state when the patient was not hypoalbuminemic;-The new steady state shows the same unbound/free drug concentration with lower bound and total drug concentrations.

## Figures and Tables

**Figure 1 antibiotics-12-00515-f001:**
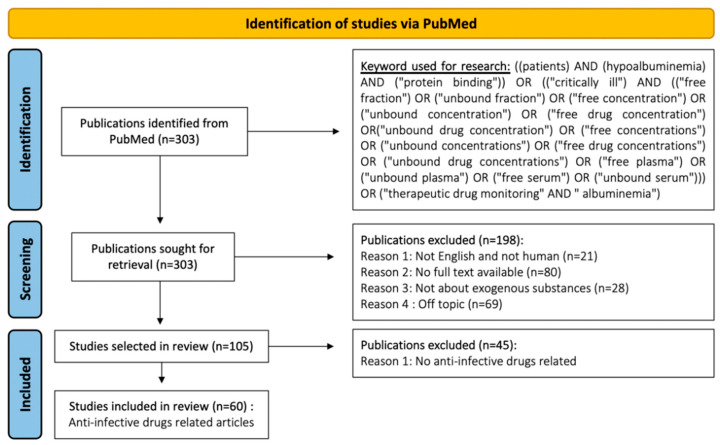
Flowchart describing PubMed review regarding unbound/free fractions/concentrations. This flowchart has been produced using the guidelines provided in [15].

**Figure 2 antibiotics-12-00515-f002:**
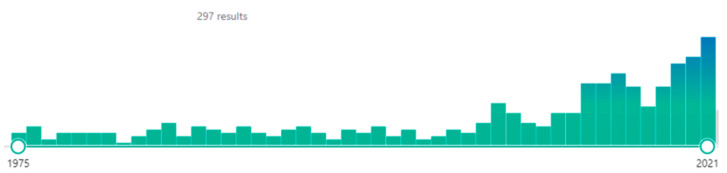
Diagram showing the articles available on PubMed when we reviewed the literature.

**Table 1 antibiotics-12-00515-t001:** Number of screened articles and articles with no incorrect sentences over the timeline.

Screening			Analysis		
Period	Number of Articles Using Keywords	Number of Included	Category 1 ^1^	Category 2 ^2^ (True + Ambigous)	At Least One False Assertion
1975–1980	18	1	0	1 (0 + 1)	0
1981–1985	15	0	0	0	0
1986–1990	19	0	0	0	0
1991–1995	17	0	0	0	0
1996–2000	15	1	0	0	1
2001–2005	16	0	0	0	0
2006–2010	32	8	1	4 (1 + 3)	4
2011–2015	61	19	2	6 (2 + 4)	13
2016–2020	81	21	3	11 (3 + 8)	10
2021	29	11	1	4 (1 + 3)	7
Total	303	61	7	26 (7 + 19)	35
%	-	-	11	43	57

^1^ Category 1—article makes no ambiguous or incorrect assertions; ^2^ Category 2—article contains ambiguous and correct assertions; %—percentages of articles in each category.

## Data Availability

Data is contained within the article or Appendix A.

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
