# Peer review of "Hypoalbuminemia and Pharmacokinetics: When the Misunderstanding of a Fundamental Concept Leads to Repeated Errors over Decades"

_antibiotics, 2023, doi:10.3390/antibiotics12030515_

Round 1

Reviewer 1 Report

Overall the authors presented an issue related to Hypoalbuminemia and Pharmacokinetics. They described incidents, frequency of incorrect/ambiguous statements/conclusion from various papers. However, instead of re-writing the statements from those papers, more value would've been added if they described the ambiguous statements and how they actually needed to be presented w.r.t to PK parameters from the respestive papers cited. Moreover, it would've been nice to cite some of the correctly published papers w.r.t Hypoalbuminemia and Pharmacokinetics and cite examples from these papers. This would give readers a better understanding of the differece in presentation of such data. 

Author Response

Overall, the authors presented an issue related to Hypoalbuminemia and Pharmacokinetics. They described incidents, frequency of incorrect/ambiguous statements/conclusion from various papers.

Comment 1

However, instead of re-writing the statements from those papers, more value would've been added if they described the ambiguous statements and how they actually needed to be presented w.r.t to PK parameters from the respective papers cited.

Answer 1

This is exactly what has been done in the supplementary material for each paper.

In the supplementary material, for each selected article passage, we indicated the type of error identified and added a specific comment.

In the manuscript, due to the length of the supplementary material, we have presented only one example per case, identifying by a set of colors what was correct (green), ambiguous (orange), and wrong (red). This is the same color scheme used in the supplementary material. In this corrected manuscript, we have added the expected wording for each selected example.

Comment 2

Moreover, it would've been nice to cite some of the correctly published papers w.r.t Hypoalbuminemia and Pharmacokinetics and cite examples from these papers. This would give readers a better understanding of the difference in presentation of such data. 

Answer 2

This is exactly what has been done in the supplementary material. In the manuscript, we have shown two examples.

Reviewer 2 Report

I thank the authors for an interesting and well written review article. The provided examples and literature data help in understanding the misconception and the resulting misinterpretation. I have a few comments and would appreciate if the authors address them.

Major comments:

1) In addition to the data presented, I would also encourage the authors to consider the lipophilicity of the drugs and include it in the discussion. Does hypo/hyperalbuminemia not affect the pharmacokinetics of lipophilic drugs?

2) Line 54 to 55 mentions, no change in the concentration of free drugs. Does that not contradict line 127-128?

3) I would appreciate the inclusion of a section discussing cases where exceptions would be seen.

Minor comments:

Please proofread the manuscript for grammatical errors.

Author Response

In addition to the data presented, I would also encourage the authors to consider the lipophilicity of the drugs and include it in the discussion. Does hypo/hyperalbuminemia not affect the pharmacokinetics of lipophilic drugs?

Answer 1 

Lipophilicity does not explain the lack of relationship between hypoalbuminemia and unbound/free concentration for drugs with a low capacity elimination, which is the subject of our article. Lipophilicity does, however, affect plasma protein binding and elimination routes.

Indeed, this article focuses on drugs that are highly bound, regardless of the factors that explain this binding.

On the other hand, the lipophilicity does not give any information on the value of the elimination capacity, as shown in the following table.

Drug

Elimination capacity

logP*

Posaconazole

low

5.5

Itraconazole

low

5.7

Bupivacaine

high

3.4

Propofol

high

3.8

Sufentanil

high

4.0

  • The logP has been documented on the following website: https://go.drugbank.com/drugs/DB01167

    Comment 2

    Line 54 to 55 mentions, no change in the concentration of free drugs. Does that not contradict line 127-128?

    Answer 2

    Line 54-55: « Indeed, when a patient presents hypoalbuminaemia, the bound concentration decreases while the unbound/free concentration remains unchanged [4–11]. »

    Line 127-128: « A possible explanation can be found in decreased levels of albumin, resulting in an increased free fraction of micafungin. »

    These two sentences do not contradict each other: the first one deals with unbound/free concentration that does not change with hypoalbuminemia while the second one is about the unbound/free fraction that changes with hypoalbuminemia because the bound concentration changes and the unbound/free concentration remains unchanged. It is precisely when readers confound these two notions (unbound/free concentration and unbound/free fraction) that the influence of hypoalbuminemia on the kinetics of the unbound/free form of the drug becomes difficult to understand. 

    Comment 3

    I would appreciate the inclusion of a section discussing cases where exceptions would be seen.

    Answer 3

    We have already specified in the manuscript the exceptions to the rules:

    Lines 15-17: « The exceptions are anesthetics highly protein-bound with high elimination capacity (i.e., < 5 drugs on the market). »

    Lines 96-97: « Even if there are few exceptions to the rule (i.e., highly bound drugs with high elimination capacities) [13],[14], there is no room to generalize these exceptions. »

     Lines 247-253: « As mentioned previously, there are few exceptions to the rule (i.e., bupivacaine, propofol, sufentanil). These drugs are highly bound drugs with high elimination capacities [13], [14]. For these drugs, which are always administered intravenously, hypoalbuminemia leads to an increase of unbound/free concentrations whereas the total concentrations remain unchanged. As there is no therapeutic drug monitoring for this kind of drugs, it is essential (i) not to generalize these exceptions and (ii) to keep in mind the correct interpretation (i.e., no impact of hypoalbuminemia on unbound/free concentrations). »

    Minor comments

    Please proofread the manuscript for grammatical errors.

    Answer 4

    Done.

Reviewer 3 Report

Gandia et.al., present literature review and explain the misconceptions about hypoalbuminemia and pharmacokinetics

Comments:

1. The point for the manuscript is well taken. However, use of equations (Well-stirred model equations perhaps) and some mathematical simulations showing when the unbound exposures don’t change and when they change upon change in plasma protein binding will be very helpful. This will supplement the video which is present in the supplementary files.

2. For the different drug cited in the different sections, please include a table citing the unbound fractions in plasma, plasma clearance, whether the drug is high extraction, low extraction or intermediate extraction ratio drug. Use experimental/clinical data along with mathematical principles will be helpful to show the point which manuscript is trying to make.

Round 2

Reviewer 2 Report

I thank the authors for addressing my comments.